# Automated retinal disease classification using deep learning and AlexNet with statistical models analysis

El-Sayed M. Elkenawy[1,2], Nima Khodadadi[3]*, Khaled Sh. Gaber[4], Ehsan Khodadadi[5], Amel Ali Alhussan[6], Doaa Sami Khafaga[6], Marwa M. Eid[7,8]

1 Department of Communications and Electronics, Delta Higher Institute of Engineering and Technology, Mansoura, Egypt, 2 Applied Science Research Center. Applied Science Private University, Amman, Jordan, 3 Department of Civil, Architectural and Environmental Engineering, University of Miami, Coral Gables, Florida, United States of America, 4 Computer Science and Intelligent Systems Research Center, Blacksburg, Virginia, United States of America, 5 Department of Chemistry and Biochemistry, University of Arkansas, Fayetteville, Arkansas, United States of America, 6 Department of Computer Sciences, College of Computer and Information Sciences, Princess Nourah bint Abdulrahman University, Riyadh, Saudi Arabia, 7 Faculty of Artificial Intelligence, Delta University for Science and Technology, Mansoura, Egypt, 8 Jadara Research Center, Jadara University, Irbid, Jordan

* Nima.khodadadi@miami.edu

## Abstract

Diabetic Retinopathy, Cataract, and Glaucoma are major retinal diseases that require early detection to prevent irreversible vision loss. This study proposes a deep learning-based framework for the automated classification of retinal images into four categories: Normal, Diabetic Retinopathy, Cataract, and Glaucoma. The dataset was compiled from publicly available retinal imaging databases, including IDRiD and HRF. Four convolutional neural network architectures—EfficientNet-B0, EfficientNet-B7, a build-from-scratch model, and AlexNet—were evaluated using multiple performance metrics. Among these, AlexNet achieved the highest overall performance, attaining an accuracy of 93.65%, sensitivity of 94.39%, specificity of 98.05%, PPV of 93.65%, NPV of 97.95%, and an F1-score of 93.74%. EfficientNet-B7 followed with an accuracy of 92.82%, confirming the strength of transfer learning in retinal feature extraction. A five-fold cross-validation further validated AlexNet's robustness, yielding a mean $R^2$ of 0.8891 with low variance, indicating consistent generalization across folds. Computational efficiency analysis showed that AlexNet achieved high diagnostic accuracy with a moderate processing time of approximately 14 minutes. Model interpretability using SHapley Additive exPlanations (SHAP) revealed that AlexNet highlighted clinically relevant retinal regions, such as the optic disc and macula, thereby enhancing transparency and clinical trust. In summary, the proposed framework demonstrates that interpretable deep learning models can deliver accurate, consistent, and explainable retinal disease classification, offering a foundation for real-time, AI-assisted ophthalmic screening systems.

**Data availability statement:** The datasets analyzed for this study can be found in https://www.kaggle.com/datasets/gunavenkatdoddi/eye-diseases-classification.

**Funding:** Princess Nourah bint Abdulrahman University Researchers Supporting Project number (PNURSP2025R754), Princess Nourah bint Abdulrahman University, Riyadh, Saudi Arabia.

**Competing interests:** The authors have declared that no competing interests exist.

# 1 Introduction

The world is experiencing hard times due to eye diseases like Diabetic Retinopathy, Cataracts, and Glaucoma, which affect millions of people, and the outcome is a severe vision handicap or blindness. According to the World Health Organization (WHO), about 2.2 billion people worldwide experience impaired vision, and a third of them have eye diseases that can be prevented or treated. The way such diseases present themselves and how they are diagnosed is very critical, given the aging society and more so as such diseases as diabetes and hypertension are becoming increasingly prevalent in the population. Consider, for example, Diabetic Retinopathy, one of the most frequent causes of blindness in the working population; the disease worsens a person's condition due to its duration and the inability to control it. This is why early detection and prevention strategies based on population-based approaches are demanded [1–3].

Another common eye ailment is cataracts, which can cause considerable reversible blindness. Cataracts are a prevalent condition, with more than 50% of people over 65 years of age having some form of cataract developing. In contrast, Glaucoma is medically referred to as the "silent boogieman of blindness" because the disease often advances slowly and without obvious signs until the advanced stage is diagnosed. Hence, given the minimal or absent presenting symptoms in many patients, prompt diagnosis and intervention are critical to avoid untreatable blindness. Prior approaches to analyzing the above conditions rely entirely on ophthalmologists who review retinal images to identify features associated with the disease. Thus, when used in such an environment, it can lead to work overload for medical personnel and be somewhat inaccurate due to its manual nature [4,5].

This research aims to evaluate the performance of multiple deep learning models applied to a dataset of retinal images comprising four classes: regular, Diabetic Retinopathy, Cataract, and Glaucoma. The chosen images are 4,200, as acquisition from different sources would ensure adequate training and testing of the models. The models under consideration include a Scratch Model, EfficientNet-B0, EfficientNet-B7, and AlexNet, all of which are employed because they have specific architectural approaches that make them naturally suited for complex image classification patterns critical to medical imaging [6,7].

Thus, this research aims to advance understanding of these models' ability to classify retinal images by systematically reviewing their performance. The specific aims are to assess the accuracy and reliability of each model and the sensitivity and specificity of the various eye diseases. Ultimately, the goal is to develop and refine methods for early detection, benefiting patients and enabling healthcare professionals to make accurate diagnoses at an early stage. The outcomes of this study will not only contribute to the understanding of the optimality of these superior models but also highlight the need to incorporate AI equipment in the facility to enhance eye treatment services.

## 2 Related works

Early and accurate diagnosis of retinal diseases is critical for preventing irreversible vision impairment and blindness. Diabetic Retinopathy (DR), Cataracts, and Glaucoma are among the top causes of visual disability in the world, and their growing occurrence following the aging population and more individuals contracting diabetes underscores the need to develop effective diagnostic technology. The profession of ophthalmology has traditionally been highly dependent on the manual analysis of retinal images- a procedure that, although clinically proven, is time-consuming, biased to the observer, and suffers due to the lack of qualified specialists. Poor infrastructure and technology have also contributed to diagnostic delays, especially in poorly resourced healthcare facilities. Nonetheless, the intersection of computer capabilities in artificial intelligence (AI), the Internet of Things (IoT), and cloud computing has led to a new era of computer-aided diagnosis (CAD), specifically through deep learning (DL) networks analyzing retinal images [8]. Some prerequisites for the diagnosis of retinal diseases include non-invasive imaging modalities such as color fundus photography and Optical Coherence Tomography (OCT). Fundus imaging enables one to detect retinal lesions, vascular defects, and optic disc abnormalities, which is particularly effective for large-scale screening [9]. Nevertheless, most past computational models have employed binary classifiers with specific pathologies; this limits their ability to generalize and achieve sound performance. Recent developments are aimed at creating multi-label and multi-class models, with optimized models such as EfficientNet-B4, and extending spatial attention blocks and focal loss functions to address class imbalance. Such architectures have also demonstrated substantial improvements in classification accuracy without a significant increase in computational complexity.

The quick development of DL has changed the medical image analysis, and in particular Convolutional Neural Networks (CNNs). CNNs learn hierarchical feature representations on their own and perform better than the conventional machine learning models that consider manually extracted features [8,10]. This has played a critical role in detecting more subtle pathological attributes in non-homogeneous datasets with varying image quality, luminance, and anatomical appearances. Hybrid algorithms combining CNNs with classical feature extraction algorithms, e.g., Histogram of Oriented Gradients (HOG) and Grey-Level Co-occurrence Matrix (GLCM), have been practical. A recent experiment using Inception V3 on RGB and green-channel images of the retina achieved an impressive F1 score of 99.39 per cent, a real demonstration of the improved predictive power of modern DL architectures [11].

At the same time, transformer-based models are becoming impressive competitors in fundus image and Optical Coherence Tomography (OCT) image classification. [12] proposed an interpretable Swin-Poly Transformer network, which can learn multi-scale retinal features by dividing the window using a shifted window and weighting the basis polynomials. Their model achieved 99.80% accuracy and 99.99% AUC on the OCT2017 and OCT-C8 datasets and also generated confidence score maps, which increased interpretability and clinical transparency. This innovation shows the increased focus on explainable AI systems in retinal diagnosis.

To further this pattern, proposed a hybrid deep learning architecture combining ResNet50 and EfficientNet-B0 [13] to classify four types of retinal diseases in OCT images. Their model achieved 97.50% accuracy by fine-tuning on large-scale OCT datasets using pre-trained ImageNet weights and alleviating overfitting by removing dense layers and adding concatenated learning objectives. Their hybrid structure enabled them to leverage the hierarchical feature extraction of ResNet and the efficient scaling strategy of EfficientNet to achieve better generalization across a variety of retinal imaging conditions.

[14] introduced MHANet, a hybrid attention network combining spatial and channel attention to enhance feature discrimination in retinopathy images. This is an attention mechanism in the dual attention model that helps the model focus on major pathology areas while reducing interference from the background. MHANet achieved classification accuracies of 96.5

[15] built upon CNN-based approaches by using a series of binary classifiers (VGG16, VGG19, ResNet50, DenseNet121, and InceptionV3) to differentiate four types of retinal diseases (CNV, DME, Drusen, and Normal) by using OCT images.

The highest accuracy, sensitivity, and specificity of their ensemble were 0.987, 0.987, and 0.996, respectively, highlighting the effectiveness of binary CNN models as strong decision-support systems by ophthalmologists.

More development in ensemble learning is described by [16], who used image preprocessing, vessel segmentation, and multi-architecture deep learning (EfficientNet-B0, VGG16, ResNet-152) to classify 11 retinal diseases. Their ensemble achieved 99.71 per cent accuracy, 98.63 per cent precision, and 99.22 per cent F-measure, demonstrating the potential of hybrid and ensemble networks to achieve high diagnostic reliability close to perfection. All these studies support the fact that the tendency toward more interpretable, hybrid, and ensemble-based models, which enhance the accuracy and transparency of retinal disease classification, is gaining momentum.

At the same time, transformer-based models are becoming increasingly effective for fundus image classification. These models can outperform traditional CNNs in multi-label classification tasks by leveraging self-attention mechanisms to capture global context. An example of such implementation has shown an improvement in AUC of more than 7% over existing state-of-the-art models, suggesting their ability to be used in large numbers in clinical diagnostics [17].

Disease-specific DR continues to be one of the most widely researched retinal diseases because it affects people worldwide and because of its mechanisms of development. According to estimates from the World Health Organization, a significant portion of global vision loss is treatable, including DR, cataracts, and glaucoma, among others [10]. Research into the epidemiology of DR has shown that rural populations and those people who have diabetes with a long history have disproportionately high rates of the disease incidence [18]. Patterns of association rule mineralization of people with diabetes have indicated that DR has a significant co-occurrence with other pathologies, including cataracts and diabetic maculopathy, which supports the need to have integrated diagnostic models that can identify more than one pathology at a time.

The studies have proved the effectiveness of the DL-based methods in the diagnosis of DR. Indicatively, U-Net architectures based on semantic segmentation have been effective in detecting red lesions at an early stage, with a sensitivity and specificity of 89% and 99%, respectively, with the overall accuracy of the system reaching above 95% identification accuracy of red lesions [19]. Ensemble learning models using decision trees also improve diagnostic reliability and achieve high performance across a range of publicly available datasets, including MESSIDOR and APTOS 2019 [20]. DL models on fundus photographs have also been used in glaucoma perception, where the morphology of the optic nerve head, especially cup-to-disc ratio, can be well detected with the help of testing the fundus images to identify the glaucomatous alterations early on, even using images that are inconclusive based on manual analysis of the optic nerve head morphology [21,22].

In order to present a brief overview of the existing research environment and its methodological heterogeneity in terms of the diagnosis of retinal disease using artificial intelligence and deep learning, Table 1 below will provide a summary of the breadth, methods, and key findings of 23 notable articles included in this review. These pieces of work cut across different disciplines, including image classification, multi-label disease detection, model optimization, epidemiological association analysis, and economic impact assessment.

## Research gap and main contribution

Although the use of deep learning in retinal disease detection has advanced significantly, several gaps remain in current research. Previous studies have been more concentrated on either fundus or OCT images alone, with most studies using single-model architectures, e.g., CNNs or EfficientNet derivatives. Although these models have shown high accuracy, they are also prone to poor generalization, especially when used on heterogeneous datasets for different retinal diseases. Moreover, current strategies tend to be computationally consuming and are therefore not applicable in real-time clinical implementation. The other major weakness is that it is not interpretable or statistically validated, which would not ensure clinical confidence or its extensive use in medical practice. In resolving these issues, this research paper will introduce a comparative model that compares several deep learning models, AlexNet, EfficientNet-B0, EfficientNet-B7, and a

**Table 1**. Summary of key literature on AI-based retinal disease diagnosis.

| Ref. | Focus Area | Methodology | Key Findings |
|---|---|---|---|
| [9] | Multi-label fundus disease classification | EfficientNet-B4 with spatial attention and focal loss | Enhanced F1-score by 2.86%; proposed error correction and improved recognition with minimal computation. |
| [10] | DL in ophthalmology | Review of CNNs for retinal image analysis | Summarized CNN-based diagnostic tools and mobile screening apps for early-stage detection. |
| [8] | DL vs. ML in medical imaging | Systematic review of 40 studies (2014–2022) | DL models outperformed ML in scalability and accuracy; emphasized need for diverse datasets. |
| [21] | Glaucoma detection via CAD systems | Morphological analysis of fundus images | Demonstrated improved accuracy and reduced inter-observer variability using DL-based CAD. |
| [18] | DR comorbidity patterns | Association rule mining and visualization | Found DR co-occurs with cataracts and maculopathy; higher incidence in rural, long-term diabetics. |
| [22] | Glaucoma classification with CNNs | DL on optic nerve head images | Validated CNN capability to detect early glaucomatous changes from fundus photos. |
| [11] | Joint detection of DR, cataracts, and glaucoma | Inception V3, HOG, GLCM, and SVM | Inception V3 reached 99.39% accuracy; confirmed green-channel image usefulness. |
| [19] | Red lesion segmentation in DR | UNet and CNN-based semantic segmentation | Accuracy 95.65%, specificity 99%, sensitivity 89%; validated on IDRiD and MESSIDOR datasets. |
| [20] | DR detection using ensemble trees | Texture and intensity features with decision trees | Achieved 94.2% accuracy and 93.51% F-measure using the APTOS 2019 dataset. |
| [17] | Multi-label classification via transformers | Transformer model with MuReD dataset | Improved AUC by 7.9–8.1% over SOTA; demonstrated superior multi-disease recognition. |
| [23] | OCT image classification using VGG-19 | Transfer learning with deep CNN | Accuracy of 99.17%; confirmed statistical validity via AUC and Cohen's kappa metrics. |
| [12] | Retinal disease classification using OCT and transformer models | Swin-Poly Transformer with interpretable architecture and polynomial cross-entropy optimization | Achieved 99.80% accuracy and 99.99% AUC; provided confidence score maps for clinical interpretability. |
| [13] | Hybrid ensemble learning for retinal OCT classification | Combined ResNet50 and EfficientNet-B0 with concatenated learning and fine-tuning strategy | Reached 97.50% accuracy with improved generalization and reduced overfitting across heterogeneous datasets. |
| [14] | Attention-driven retinal disease recognition | MHANet integrating spatial and channel attention mechanisms | Achieved 96.5% and 99.76% accuracy on two public OCT datasets; improved focus on pathological regions. |
| [15] | Binary CNN-based multi-class retinal disease classification | Four binary CNN classifiers using VGG16, VGG19, ResNet50, DenseNet121, and InceptionV3 | Reported 0.987 accuracy, 0.987 sensitivity, and 0.996 specificity on OCT images. |
| [16] | Ensemble-based segmentation and classification of retinal diseases | Adaptive Gaussian kernel preprocessing with EfficientNet-B0, VGG16, and ResNet-152 ensemble | Classified 11 retinal diseases with 99.71% accuracy, 98.63% precision, and 99.22% F-measure. |

modified CNN, for the automated classification of retinal diseases. The proposed dataset is balanced and multi-source, covering four classes: Normal, Diabetic Retinopathy, Cataract, and Glaucoma. Both models are evaluated using extensive performance metrics, including accuracy, sensitivity, specificity, positive predictive value (PPV), negative predictive value (NPV), and F1-score. The robustness of the results is ensured by the use of a nonparametric test in the statistical analysis. Results show that AlexNet is more stable and efficient than other neural networks, achieving higher performance while maintaining computational efficiency. Generally, the research provides a comprehensive, portable framework for assessing deep learning models for retinal disease classification. It supports the development of deployable, explainable, and resource-efficient AI systems in ophthalmology.

To sum up, the literature shows increasing agreement on the transformative potential of deep learning for the diagnosis and treatment of retinal diseases. In the course of research development, the creation of clinically sound, interpretable, and fair AI solutions that can be easily incorporated into the ophthalmic practice should be a priority. The combination of AI innovation and medical knowledge is essential to addressing the rising burden of visual impairment worldwide and to providing accessible, precise, and timely eye care to those who need it.

## 3 Material and methods

The general flow of the suggested automated retinal disease classification system is depicted in Fig 1. The architecture starts with the retrieval of retinal fundus images from the diabetic retinopathy dataset and an extensive data preprocessing stage, including image resizing, normalization, and data augmentation to improve model generalization. The dataset is divided into two parts after preprocessing into training and testing sets with a 80:20 ratio to balance out the evaluation process. The four baseline deep learning models, which are CNN, EfficientNet-B0, EfficientNet-B7, and AlexNet, are subsequently trained and optimized to label the retinal images in different classes. The model's performance is then checked using statistical analysis to ensure it is reliable and significant. The systematic working plan will ensure that every step of the procedure leads to correct, interpretable, and statistically validated results for retinal disease diagnosis.

### 3.1 Dataset description

The data this study will use include 4,200 retinal images from various sources, including IDRiD, Oculus Recognition, and HRF. It is divided into four classes: Normal, Diabetic Retinopathy, Cataract, and Glaucoma. Therefore, it encompasses almost all common eye diseases. The quality of the images also varies in terms of resolution, color contrast, lighting, and picture realism, which is desirable and consistent with the conditions in which the diagnosticians work. This enables the models to learn and generalize across a broad spectrum of real-world situations, and the classification is strong. The variety of images per disease type has enough training and testing images, and they capture and isolate the distinguishing aspect of each disease type. A visual sample of the dataset is shown in Fig 2, demonstrating representative images of each of the four classes: Normal/Suspect, Diabetic Retinopathy, Cataract, and Glaucoma. This dataset is essential for training and testing of deep learning models in retinal disease diagnosis.

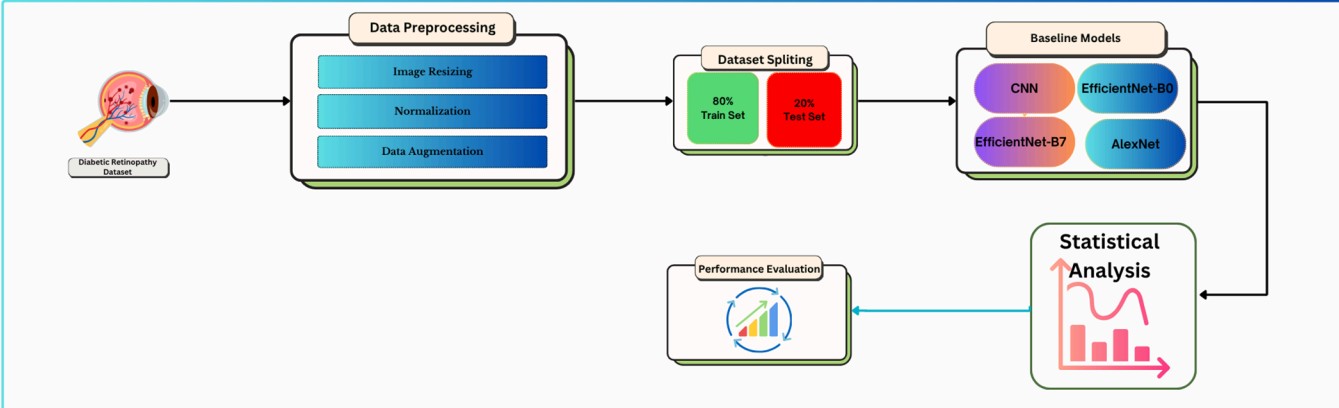

**Fig 1**. **Proposed workflow of the automated retinal disease classification system.**

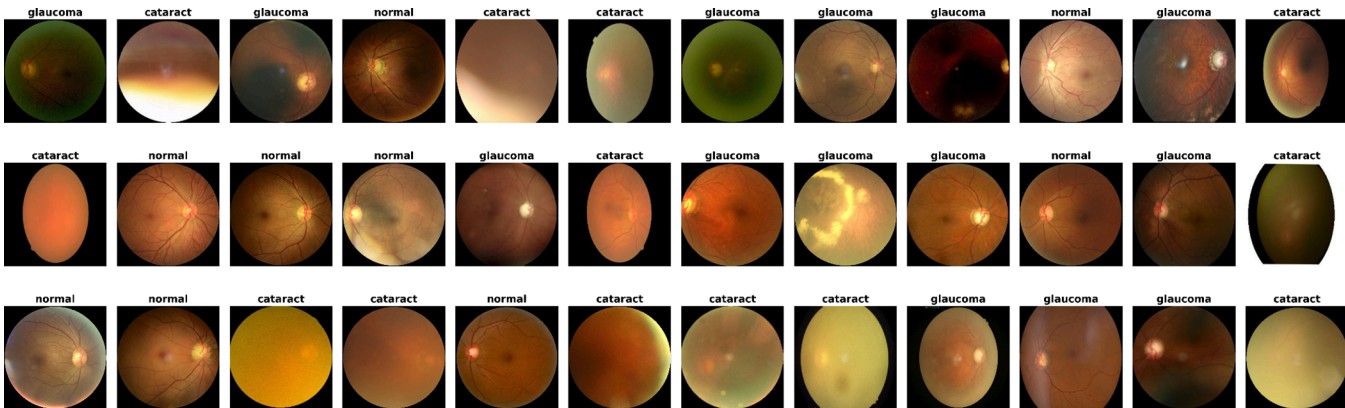

**Fig 2**. Sample from the dataset with all classes.

### 3.2 Data preprocessing

Before inputting the retinal images into the deep learning models, several preprocessing steps were performed to ensure the images were suitable for model training. Initially, all images were resized to a uniform dimension of 224 × 224 pixels. This resizing minimizes variability and is optimal for the deep learning models employed in this research. Subsequently, the images were converted to tensors, and their pixel values were rescaled to `float32` within the normalized range of 0 to 1, making them more compatible with neural network processing. The normalization process then adjusted the pixel intensities using means of approximately 0.485, 0.456, and 0.406, with corresponding standard deviations of roughly 0.229, 0.224, and 0.225. This normalization ensures that the data distribution has a zero mean and unit variance, improving learning efficiency and accelerating convergence during training. Additionally, three fundamental spatial transformations were applied to augment the dataset, thereby enhancing the model's ability to generalize and detect diverse retinal conditions.

### Comparative analysis of deep learning models

The models utilized in this paper, such as CNN, EfficientNet-B0, EfficientNet-B7, and AlexNet, possess distinct features that affect their performance in classifying retinal diseases, including Normal, Diabetic Retinopathy, Cataract, and Glaucoma.

The CNN is a comparably lightweight and efficient architecture inspired by the VGG architecture and can be applied in settings with limited computational resources. It can achieve acceptable performance, especially with well-preprocessed datasets, thanks to its simple design and robust feature-extraction layers. Nevertheless, its limited depth and representational power may not be adequate to depict more subtle pathological characteristics as well as more profound architectures.

EfficientNet-B0 is the most efficient and accurate model due to compound scaling and depthwise separable convolutions. It has a relatively low computational cost and performs better than traditional CNNs; thus, it can be used in real-time applications. The architectural depth and Swish activation also enable it to learn discriminatory features applicable to ophthalmological diagnostics with shorter inference times.

The newest and most resource-intensive EfficientNet-B7 model offers the best-in-the-business classification results. Its ability to match complex visual patterns makes it particularly well-suited for recognizing subtle variations in categories of retinal diseases. However, this great precision comes at the expense of higher resource requirements, so it is more suited to offline analysis or high-performance computing systems.

Although AlexNet was a groundbreaking model in deep learning, it has since been surpassed by more recent models. It has strong baseline performance thanks to its ReLU activation and dropout regularization. Nevertheless, it is less efficient at extracting features and achieving classification accuracy, particularly when using high-resolution retinal images.

### System configuration and training setup

All experiments, including model training, validation, and testing, were conducted on a high-performance local workstation to ensure computational efficiency, scalability, and reproducibility of results. The detailed hardware configuration employed throughout both the training and evaluation phases is presented in Table 2. These specifications are particularly important in deep learning research, as they directly influence model convergence speed, the feasible network complexity, and the ability to process high-resolution retinal images effectively.

This configuration ensured that all models were trained efficiently under consistent experimental conditions. The combination of a high-core-count CPU, a modern NVIDIA GPU architecture, and high-speed DDR5 memory provided optimal throughput for data-intensive deep learning workloads, ensuring reproducibility of results across multiple runs.

Each deep learning architecture—Scratch Model, EfficientNet-B0, EfficientNet-B7, and AlexNet—was implemented and trained using the PyTorch framework. All models were optimized under the same hyperparameter configuration to ensure a fair and unbiased comparison. The key optimization and training parameters are summarized in Table 3.

Training was performed using the hardware detailed above. The average training time per model varied with architectural depth, with AlexNet taking approximately 14 minutes, EfficientNet-B0 14.7 minutes, and EfficientNet-B7 20 minutes. To enhance model convergence and prevent overfitting, early stopping and learning rate scheduling were used throughout training. This consistent configuration ensured an equitable and reproducible experimental environment across all architectures.

### 3.3 Metrics descriptions

**Accuracy:** Quantifies how many of the total predictions were correct; both true positive and actual negative cases were considered. It is a global performance measure that is not always appropriate, especially when dealing with imbalanced data [24].

$$Accuracy = \frac{TP + TN}{TP + TN + FP + FN}$$

**Sensitivity (True positive rate):** Specificity refers to the power of a model to identify negatives. A higher TNR means fewer false negatives, crucial in medical diagnosis since failure to diagnose a positive case can lead to fatal effects [24].

$$Sensitivity = \frac{TP}{TP + FN}$$

Table 2. **Complete system specifications used for model training and evaluation.**

| Component | Specification |
|---|---|
| CPU | Intel Core i7-14700K (20 Cores, 28 Threads, up to 5.6 GHz) |
| RAM | 64 GB DDR5 (5600 MHz) – Dual Channel (2×32 GB) |
| GPU | NVIDIA GeForce RTX 4070 Ti Super (16 GB GDDR6X VRAM) |
| Motherboard | Intel Z790 Chipset with PCIe 4.0/5.0 Support |
| Primary Storage | 2 TB NVMe SSD (PCIe 4.0) – Samsung 990 PRO |
| Secondary Storage | 4 TB HDD (7200 RPM) – Western Digital Blue |
| Power Supply Unit (PSU) | 850 W 80+ Gold Modular – Corsair RM850x |
| Cooling System | 280 mm AIO Liquid Cooler – Corsair iCUE H115i |
| Chassis | Fractal Design Meshify 2 – Mid-Tower High-Airflow Case |
| Operating System | Windows 11 Pro (64-bit) |

**Table 3**. Training and optimization parameters for all deep learning models.

| Parameter | Configuration |
|---|---|
| Deep Learning Framework | PyTorch 2.2.2 (Python 3.11) with CUDA 12.1 and cuDNN 8.9 |
| Optimizer | Adam Optimizer |
| Learning Rate | 0.0001 (decayed by factor of 0.1 every 10 epochs) |
| Batch Size | 32 |
| Number of Epochs | 15 |
| Loss Function | CrossEntropyLoss (categorical) |
| Weight Initialization | Kaiming (He) Normal Initialization |
| Regularization | Dropout (rate = 0.5) and L2 weight decay ($1 \times 10^{-4}$) |
| Data Augmentation | Random rotation ($\pm15°$), horizontal/vertical flips, and brightness adjustment |
| Input Image Resolution | 224 × 224 pixels |
| Learning Rate Scheduler | StepLR with step size = 10 and gamma = 0.1 |
| Early Stopping | Patience = 10 epochs (monitored on validation loss) |

**Specificity (True negative rate):** Specificity measures the proportion of true negatives among the total number of negatives the test yields. It is helpful because it helps distinguish between classes when the price the system pays for a mistake is high [24].

$$\text{Specificity} = \frac{TN}{TN + FP}$$

**Positive Predictive Value (PPV):** PPV stands for positive predictive value, which is the accuracy of positive predictions. Its calculation assesses the reliability of the model's positive-class predictions [25].

$$\text{PPV} = \frac{TP}{TP + FP}$$

**Negative Predictive Value (NPV):** NPV signifies the ratio of correct pessimistic predictions. It is beneficial when the negative data class is more important [25].

$$\text{NPV} = \frac{TN}{TN + FN}$$

**F-Score:** The F-Score is a measure that combines precision and recall in a way that serves as a compromise between sensitivity and positive predictive value. It is beneficial if the classes are uneven [25].

$$\text{F Score} = 2 \times \frac{\text{Precision} \times \text{Recall}}{\text{Precision} + \text{Recall}}$$

## 4 Experimental results

The results of the different deep learning models used in this study were compared through some of their key metrics, which include: precision (accuracy), recall (sensitivity), recall (true positive fraction), specificity (true negative fraction), positive predictive value (PPV), negative predictive value (NPV), and F-measure. All these evaluation criteria provide a complete picture of each model's ability to appropriately classify retinal images into four diagnostic categories: Normal, Diabetic Retinopathy, Cataract, and Glaucoma.

In addition to these standard assessment criteria, computational efficiency was another important performance measure added to this study, as the average training and inference times (in minutes) for each model were recorded. Temporal efficiency is an assessment that would provide a more comprehensive view of model performance, especially in clinical settings where rapid diagnostic assistance is needed.

As depicted in Table 4 model comparison, the overall accuracy (0.9365) and specificity (0.9805) of AlexNet proved to be the best, which is the capability to find normal retinal conditions with few false positives. EfficientNet-B7 was close by with an accuracy of 0.9282 but with a moderate computation cost of 20.13 minutes, which represents a good trade-off in performance and efficiency. EfficientNet-B0 was slightly less accurate (0.9133) but faster to converge and more cost-efficient in computation, highlighting the advantage of lightweight architectures for scalable deployment. Conversely, the scratch model yielded lower scores across all measures, further confirming the effectiveness of transfer learning and initial weights in enhancing feature generalization in retinal disease classification.

Table 4 in the appendix highlights the relative efficiency of these architectures and the design decisions that are specific to them and affect their predictive performance. Combined, the incorporation of diagnostic accuracy, sensitivity, specificity, and time efficiency demonstrates that deeper, pre-trained models, including EfficientNet and AlexNet, not only increase the accuracy and interpretability of retinal disease classification but also ensure that their computational capabilities remain within real-time clinical use.

Fig 3 is a visual summary of an evaluation of metrics on all deep learning architectures used in this work, including the Scratch Model, EfficientNet-B0, EfficientNet-B7, and AlexNet. This heatmap summarizes six key performance measures, including Accuracy, Sensitivity (True Positive rate), Specificity (True Negative rate), Positive Predictive Value (PPV), Negative Predictive Value (NPV), and F-score, in an attempt to offer a multidimensional view of the diagnostic performance of each model.

The color change from blue (poor performance) to red (good performance) allows easy visual distinction among architectures and emphasizes their differences. As noted, EfficientNet-B7 and AlexNet show better overall consistency across all measures, particularly high specificity and NPV, which indicate their ability to reduce false negatives and false positives simultaneously. On the other hand, the scratch model is on the back foot in terms of precision metrics, and transfer learning plays a significant role in feature extraction for retinal image classification.

This comparative visualization enables an intuitive understanding of model performance. It shows that pre-trained and hybrid networks perform better across most metric dimensions than baseline networks, supporting their use in clinical-grade diagnostic systems.

Fig 4 below (processing time) demonstrates the relative processing speed of the four deep learning models, Scratch Model, EfficientNet-B0, EfficientNet-B7, and AlexNet, that are employed in this project. This value provides an empirical description of the computing efficiency of each model, in minutes, for both the training and inference processes. Processing time analysis is essential for giving insights into the real-world deployability of these architectures in the clinical setting, where fast diagnostic inference is frequently needed. As presented, EfficientNet-B7 has the longest processing time due to its greater model depth and number of parameters, while AlexNet has the shortest time, reflecting its efficiency and low computational complexity. These results highlight the trade-off between model accuracy and computational requirements between architectures.

To further confirm the strength and trustworthiness of the AlexNet model, a 5-fold cross-validation was performed. This statistical validation method provides an overall evaluation of the model's generalization ability by training and testing it across multiple data partitions. This will reduce bias that can arise from a single training-test split and provide a more objective assessment of predictive performance.

**Table 4**. **Performance comparison of different deep learning models for retinal disease classification.**

| Model | Accuracy | Sensitivity (TPR) | Specificity (TNR) | PPV | NPV | F-Score | Time (min) |
|---|---|---|---|---|---|---|---|
| Scratch Model | 0.8466 | 0.8601 | 0.9542 | 0.9506 | 0.8409 | 0.8466 | 14.85 |
| EfficientNet-B0 | 0.9133 | 0.9195 | 0.9723 | 0.9716 | 0.9138 | 0.9138 | 14.69 |
| EfficientNet-B7 | 0.9282 | 0.9293 | 0.9766 | 0.9764 | 0.9284 | 0.9284 | 20.13 |
| AlexNet | 0.9365 | 0.9440 | 0.9805 | 0.9795 | 0.9374 | 0.9365 | 14.09 |

## Seaborn Heatmap with Annotations for Metric Comparisons Across Models

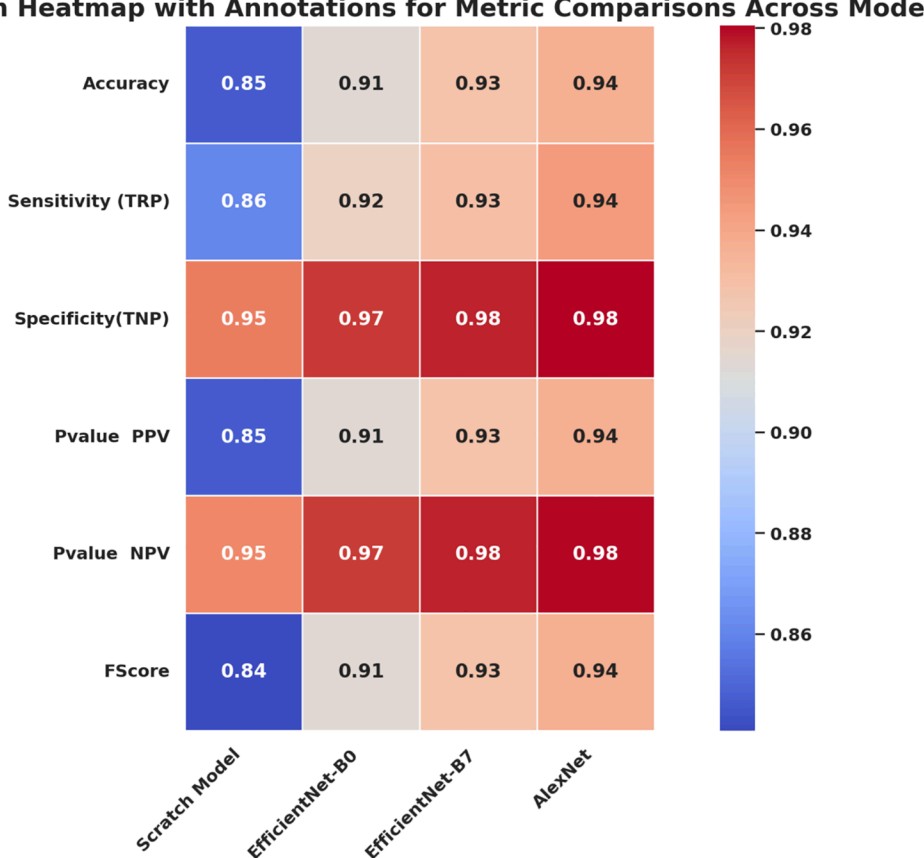

**Fig 3. Heatmap comparison of performance metrics across the four evaluated models: Scratch Model, EfficientNet-B0, EfficientNet-B7, and AlexNet.**

Table 5 shows all the detailed results of this experiment, with the coefficient of determination of a particular fold (($R^2$)) and the associated summary statistics. The fold-specific values of($R^2$) are between 0.8647 and 0.9090, with an average value of ($R^2$) 0.8891 and a standard deviation of 0.0160. These findings indicate a slight difference in performance, with the model showing consistent predictive behavior across multiple runs. The interquartile range (Q25-Q75: 0.8870-0.8949) indicates that the majority of fold performances are highly concentrated around the mean, suggesting that learning dynamics are stable and generalization is effective.

The current vertical presentation provides a clearer picture of the fold-wise behavior of the model and enhances the statistical credibility of the training process. The low variance across folds indicates that AlexNet does not overfit to small data sets, and its results can be replicated across different sample distributions. Clinically, this type of stability is vital because it ensures the model's diagnostic ability can be safely applied to invisible patient images. In general, these cross-validation findings reinforce the empirical evidence of AlexNet's effectiveness, performance, and suitability for deployment in automated retinal disease classification processes.

The model shows high stability in performance, as demonstrated in Fig 5. This number is a graphical indicator of the $R^2$ scores for each validation cycle of the AlexNet architecture, showing the consistency and reliability of the architecture's predictive characteristics. The relatively stable distribution of the bars suggests that the model performs similarly across all folds, with the slight deviations reflecting natural data heterogeneity rather than unstable learning.

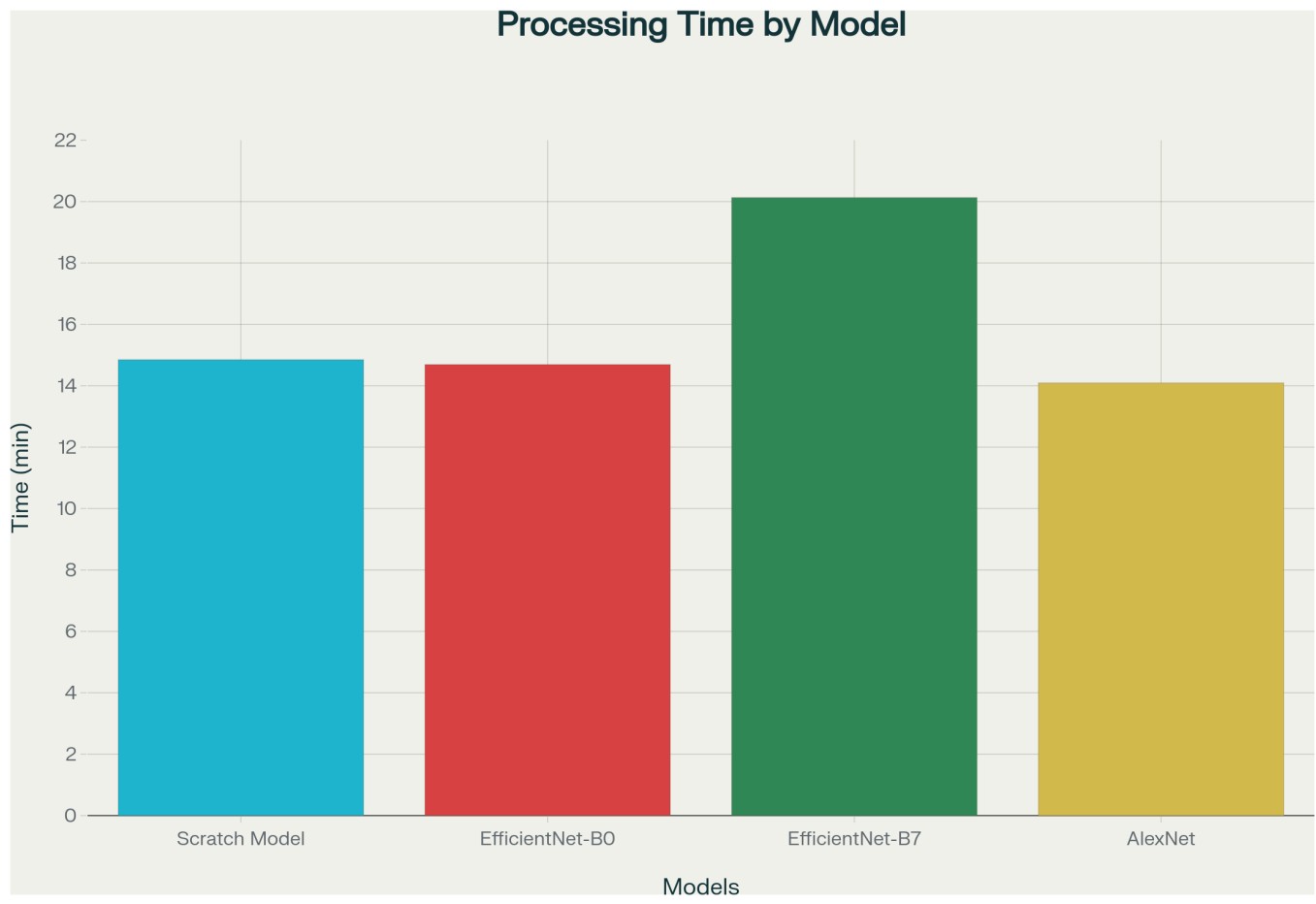

**Fig 4**. **Processing time comparison for the scratch model, EfficientNet-B0, EfficientNet-B7, and AlexNet.**

**Table 5**. **Five-fold cross-validation results for AlexNet.**

| Metric | Value |
|---|---|
| Fold_1_R2 | 0.8899464607 |
| Fold_2_R2 | 0.8870289922 |
| Fold_3_R2 | 0.8647268414 |
| Fold_4_R2 | 0.8949407935 |
| Fold_5_R2 | 0.9089913964 |
| Mean_R2 | 0.8891268969 |
| Std_R2 | 0.0160377315 |
| Min_R2 | 0.8647268414 |
| Max_R2 | 0.9089913964 |
| Median_R2 | 0.8899464607 |
| Q25_R2 | 0.8870289922 |
| Q75_R2 | 0.8949407935 |

The use of such visual analysis supports the numerical results offered in Table 5 of the article (see Table 2): the results suggest that AlexNet does exhibit high levels of generalization. The fact that the fold differences are smaller than in Fold 5 and that the difference in Fold 3 is slightly higher, and the opposite, also demonstrates that the model has satisfactory

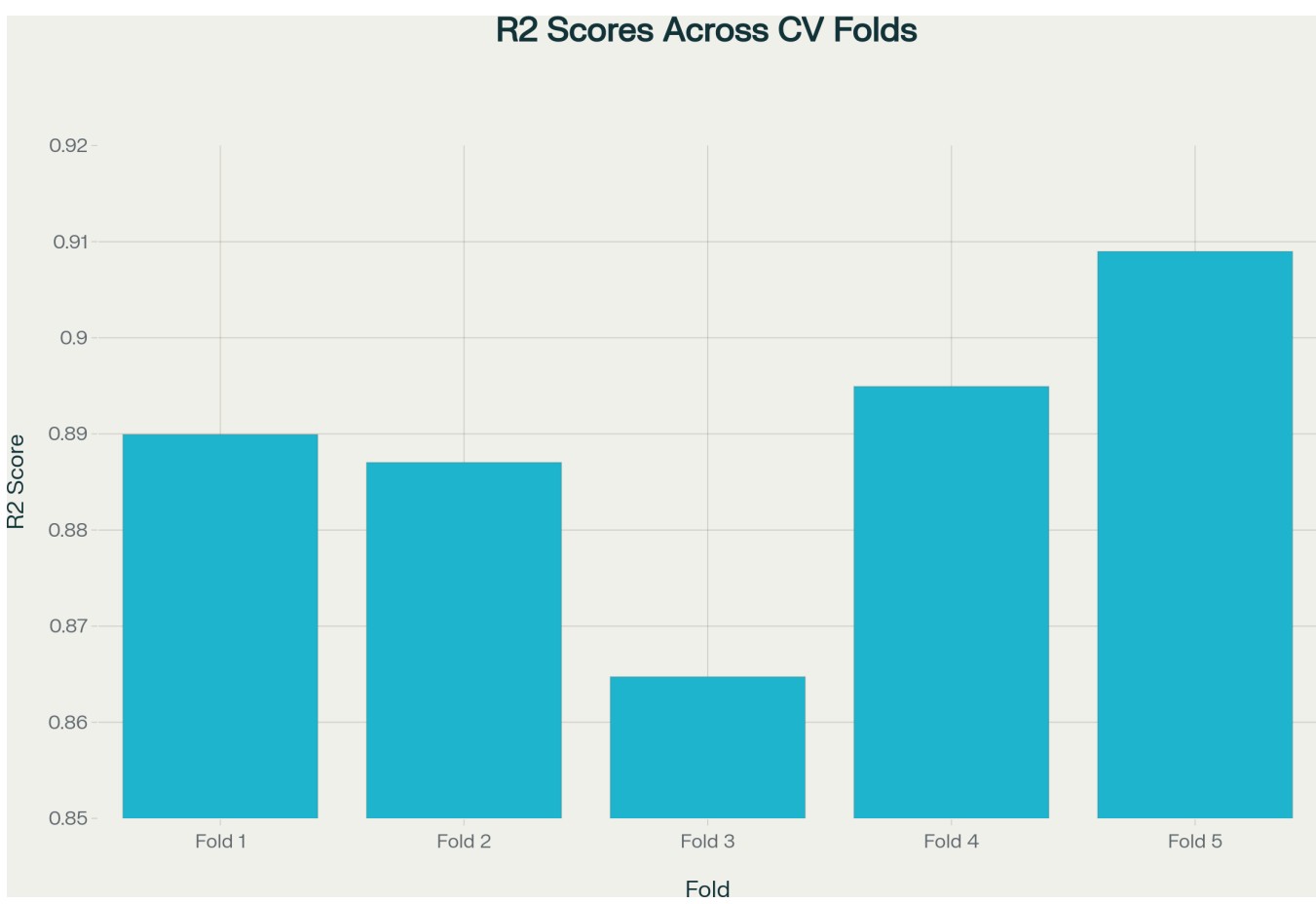

**Fig 5**. Bar chart showing $R^2$ scores across five cross-validation folds for the AlexNet model.

performance even when the training and testing splits are changed. The consistency supports the assertion that the architecture can achieve reproducible predictive accuracy across a wide range of subsets of the retinal image dataset, a critical requirement for reliable clinical deployment.

The performance of each deep learning model was evaluated using various metrics derived from confusion matrices and training plots. The effectiveness of each model in classifying retinal images into four categories—Normal, Diabetic Retinopathy, Cataract, and Glaucoma—was assessed, and detailed results are presented below.

The radar chart in Fig 6 shows the six benchmark indicators of the Scratch Model and the four architectures: EfficientNet-B0, EfficientNet-B7, and AlexNet. Table 4 presents the TNR, Sensitivity (TPR), accuracy, F-Score, NPV, and PPV for each model, providing an overall idea of their performance and limitations. Notably, EfficientNet-B7 and AlexNet achieve superior and stable performance across all tested metrics, with EfficientNet-B0 slightly behind. This comparison has revealed that EfficientNet-B7 and AlexNet offer the best compromise across these measures for this classification task.

Thus, the detailed residual analysis was performed to evaluate the model fit and the nature of errors as shown in Fig 7. This group of plots includes some essential elements: First, we have the Residual Plot, a plot of residuals versus the predicted values of Y. In an ideal situation, the coordinates of this scatter plot should cluster around the graph median at (0,0), since there should be no bias in the model's predictions. Then the plot of homoscedasticity is presented,

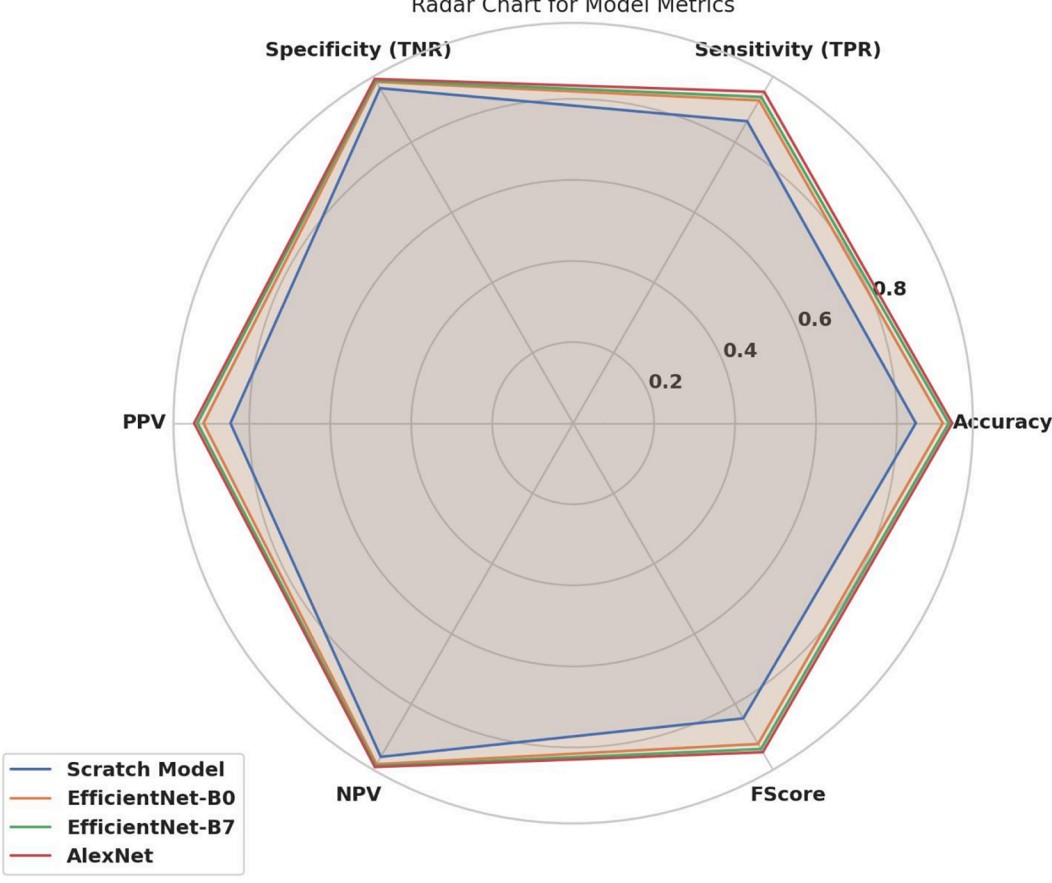

**Fig 6**. Radar chart of model metrics.

in which the absolute residuals are compared to the predicted values, and a reasonable level of homogeneous variability is observed. This plot is an important part of regression analysis that allows testing and comparing the variability of the residuals at all levels of the selected prediction. In addition, the QQ Plot shows that the special residuals are not very different from the normality curve required for further statistical analysis. Finally, but not least, a residual heat map shows the distribution of residual values for each model and may be particularly useful for giving a general idea of the distribution of prediction errors across levels of predicted values.

Aside from the residual analysis, Fig 8 shows the frequency distribution of accuracy scores of the tested models. More importantly, the Scratch Model has a lower mean accuracy and higher variance, indicating that its predictions are less accurate. On the other hand, models like EfficientNet-B0, EfficientNet-B7, and AlexNet achieve higher and more centralized accuracy, suggesting greater precision in performance. Specifically, EfficientNet-B7 and AlexNet show stable performance, with higher accuracy. This distribution of accuracy scores illustrates the performance of EfficientNet-B7 and AlexNet, justifying the selection of these two models for the subsequent analysis. The stability and accuracy of these models ensure their effective enhancement and implementation, as well as their application in real-world projects.

Fig 9 shows the interpretation analysis of the retinal disease classification model presented in AlexNet using SHapley Additive exPlanations (SHAP). This visualization can provide a detailed analysis of the roles of the various parts of the input fundus images in the model's decision-making process. In the left column, the original retinal images are shown, and the SHAP explanation maps in the right column correspond to them. The color scale runs from blue to red, with blue areas

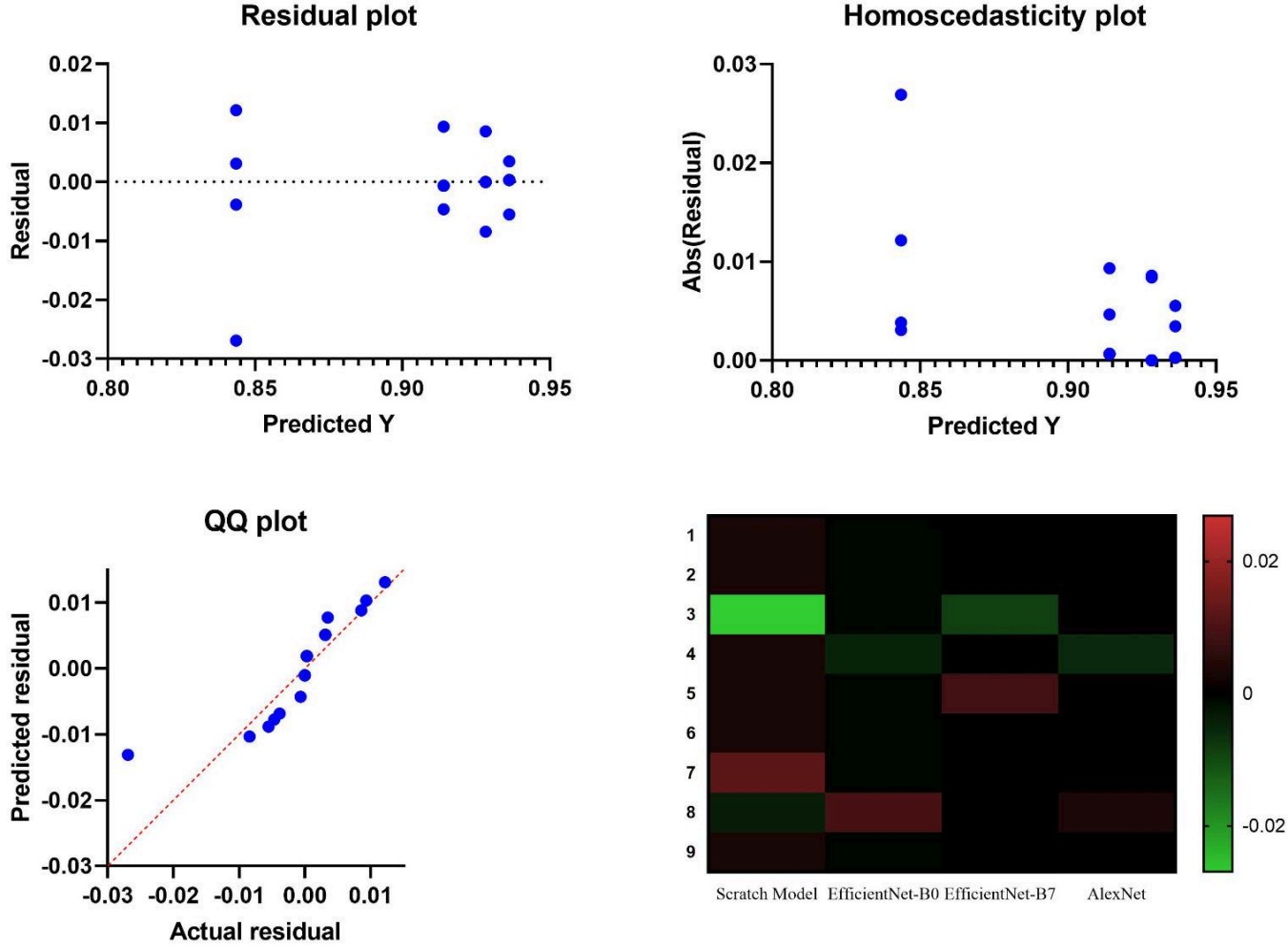

**Fig 7**. **Assessment of the accuracy of proposed and compared models.**

representing negative SHAP values (features that are working against the prediction) and red areas representing positive SHAP values (features that are working toward the predicted class).

Based on the analysis of these explanations, one can note that the model centers on clinically relevant retinal areas, including the optic disc, the macula, and the vascular structures, when generating predictions. Such agreement of the model attention and anatomical relevance increases the interpretability and clinical reliability of the system. The SHAP maps indicate that the AlexNet classifier distinguishes between pathology and normal retinal structures with reasonably high accuracy, thereby corroborating the reliability of the diagnosis.

Altogether, this statistic highlights the need to implement explainable artificial intelligence (XAI) methods into the diagnostic workflow, so that the model's high performance is complemented by clear, understandable decision-making, which is an important factor in the clinical implementation of ophthalmology.

Fig 10, the confusion matrix, shows the classification accuracy of the AlexNet model on the four retinal disease classes, namely, Normal, Diabetic Retinopathy, Cataract, and Glaucoma. The matrix is a more detailed table of actual/hypothesized labels that can be used to produce a more fine-grained evaluation of the model's reliability, in addition to overall accuracy measures. The x and y axes represent correctly and incorrectly classified samples, respectively.

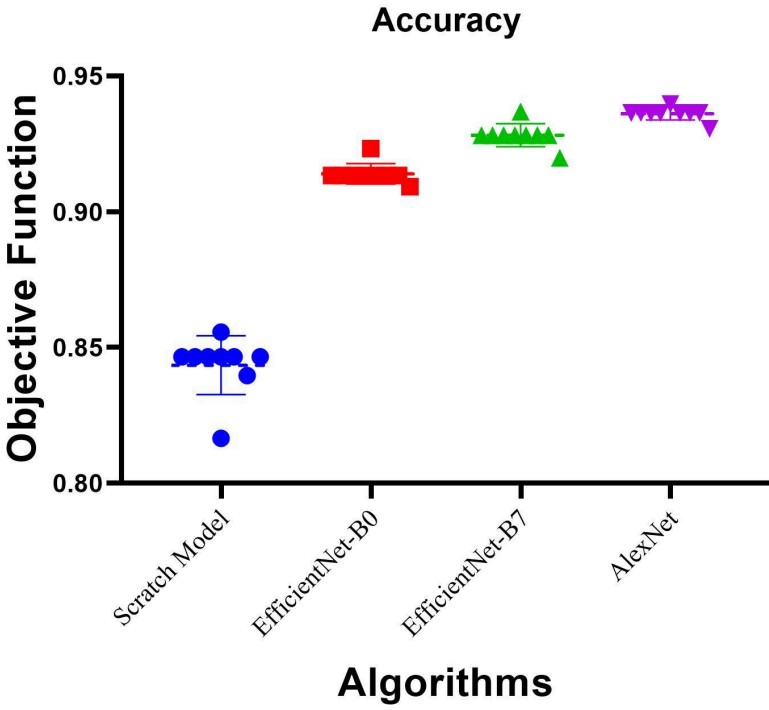

**Fig 8**. **Residual values and heatmap analysis for models.**

As shown, the AlexNet model exhibits strong diagonal concentration, indicating high classification accuracy and low confusion among classes. It is worth noting that the images of Normal and Diabetic Retinopathy depict almost perfect classification. In contrast, slight misclassification is observed between the Cataract and Glaucoma classes, which can be explained by slight visual overlap in retinal appearance. These differences are inevitably difficult, even for clinical specialists, and distinguishing retinal diseases is particularly challenging.

This confusion table, therefore, supports the quantitative performance indicators presented in the previous section, indicating that AlexNet can successfully differentiate between pathological and non-pathological retinal conditions. Its accuracy in multi-class classification is an important attribute, indicating that it is well-suited as a strong diagnostic aid in ophthalmic screening. Table 6 compares performance regarding various parameters like accuracy, loss, and others between the Scratch Model, EfficientNet-B0, EfficientNet-B7, and AlexNet. The theoretical and actual median accuracy elements are presented in the Table for the two models. The theoretical median remains constant at zero across all models, but the actual median accuracies differ: Scratch Model at 0.8466, EfficientNet-B0 at 0.9133, EfficientNet-B7 at 0.9282, and AlexNet at 0.9365, all tested at nine values each. The comparison presented above shows differences in accuracy level, but AlexNet has the highest median value.

Table 7 shows the Wilcoxon signed rank test analysis of various classification models: the Scratch Model, EfficientNet-B0, EfficientNet-B7, and AlexNet. The $w$ value across all models was 45, and the total positive ranks were also 45, reflecting a clear positive sentiment across all comparisons. However, the P value for each model was 0.0039, which is statistically significant because it is well below the accepted alpha level of 0.05. All models were deemed to provide meaningful results, as all reported performance differences were significant. The calculated values range from 0.8466 to 0.9365; therefore, the difference in errors between the models is comparatively small. In conclusion, the results obtained in the examined activities demonstrate the adequacy of the classification models under consideration.

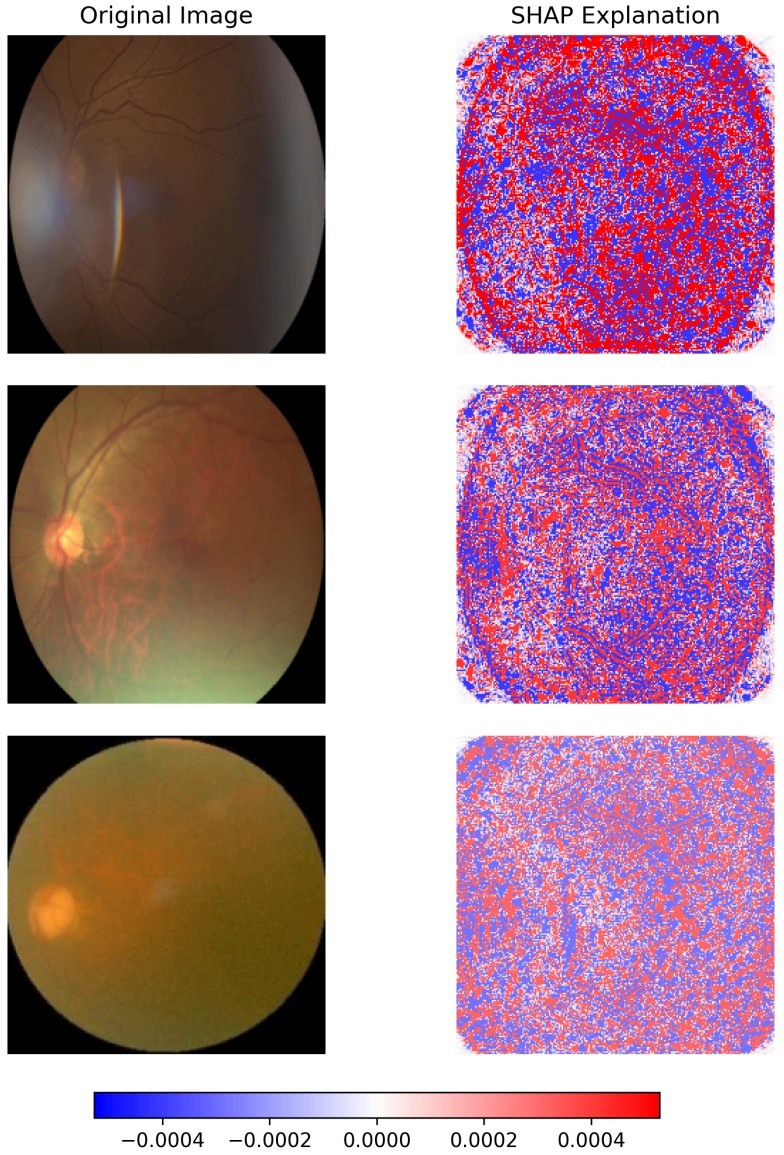

**Fig 9**. **Visualization of SHAP-based interpretability analysis for three representative retinal images.**

## 5 Discussion

The relative analysis of the four deep learning architectures, including the Scratch Model, EfficientNet-B0, EfficientNet-B7, and AlexNet, revealed that there was a large difference in their performance, computation efficiency, and interpretability. AlexNet achieved the best overall diagnostic accuracy (93.65%), high sensitivity (94.39%), and specificity (98.05%), demonstrating the strength of this algorithm in distinguishing between Normal, Diabetic Retinopathy, Cataract, and Glaucoma. The findings emphasize that conventional architectures, when properly trained, may still outperform more advanced and intricate architectures in particular situations in medical imaging. The excellent results of AlexNet can be explained by its balanced architectural depth, effective convolutional operations, and well-developed transfer learning capabilities. Although EfficientNet-B7 was a competitive model in terms of accuracy (92.82%), it has a longer processing

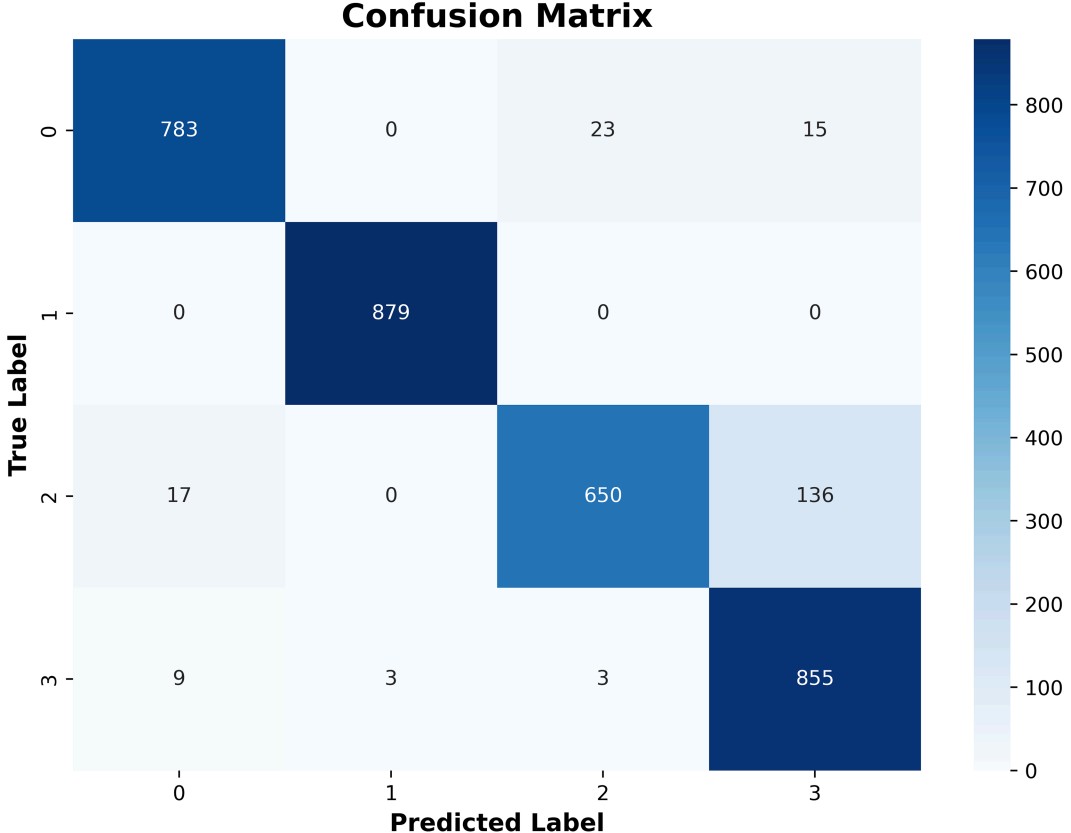

**Fig 10**. **Confusion matrix of the AlexNet model showing classification performance.**

**Table 6**. **Model performance summary.**

| Metric | CNN | Eff-B0 | Eff-B7 | AlexNet |
|---|---|---|---|---|
| Theor Median | 0 | 0 | 0 | 0 |
| Actual Median | 0.8466 | 0.9133 | 0.9282 | 0.9365 |
| Num of Values | 9 | 9 | 9 | 9 |

time (around 20 minutes) and a high computational cost, highlighting the trade-off between model complexity and inference efficiency. Although EfficientNet-B0 and the scratch model were faster, they were less predictive and less stable, indicating that moderately deep pre-trained networks are better suited to clinical settings that demand both speed and reliability. Cross-validation results further supported these observations. The mean standard deviation of 0.0160 indicates that the minimum variance across folds was achieved by AlexNet. This statistical consistency means the model generalizes well across subsets of data, so it is less likely to overfit. This is crucial in medical imaging, where there is high inter-patient variation, and the robustness of the model determines its clinical applicability. Besides quantitative performance, interpretability was also a major issue in evaluating clinical readiness. The analysis of SHapley Additive exPlanations (SHAP) indicates that the AlexNet model focuses its attention on parts of the anatomy that could be considered significant for base predictions, including the optic disc and macular regions. The correspondence between model attention and expert-relevant features increases clinical trust, thereby meeting a key criterion for explainable AI (XAI) in healthcare. This combination of SHAP analysis thus bridges the gap between algorithmic accuracy and clinical interpretation of the output, making the model's decision-making process transparent and verifiable. These findings were supported by the

**Table 7. Wilcoxon signed rank test for classification models.**

| Scratch Model | EfficientNet-B0 | EfficientNet-B7 | AlexNet | |
|---|---|---|---|---|
| Sum of signed ranks (W) | 45 | 45 | 45 | 45 |
| Sum of positive ranks | 45 | 45 | 45 | 45 |
| Sum of negative ranks | 0 | 0 | 0 | 0 |
| P value (two tailed) | 0.0039 | 0.0039 | 0.0039 | 0.0039 |
| Exact or estimate? | Exact | Exact | Exact | Exact |
| P value summary | ** | ** | ** | ** |
| Significant | Yes | Yes | Yes | Yes |
| Discrepancy | 0.8466 | 0.9133 | 0.9282 | 0.9365 |

confusion matrix analysis, which plotted the exact decision boundaries of the model. Most of the samples have been adequately identified, and there has been minor misclassification between the Cataract and Glaucoma categories - understandably, these two categories have similar viewing features. These smaller inconsistencies indicate future opportunities to enhance them through domain-specific data augmentation or multimodal integration of imaging, e.g., by combining fundus and OCT data and increasing feature discriminability. Computationally, the system setup used, which consisted of an Intel i7-14700K CPU, NVIDIA RTX 4070 Ti Super GPU, and 64 GB of DDR5 memory, was efficient in parallel processing and fast convergence. The comparison of processor time across models also showed that an efficient deployment of hardware could have a large impact on both training and inference accuracy, and it would be worthwhile to focus on scalable, resource-aware model deployment strategies. In general, the paper has shown that moderately deep CNN networks such as AlexNet can achieve an effective balance between accuracy, interpretability, and computational efficiency. The results support the practical use of interpretable deep learning systems by integrating them into the ophthalmic workflow, especially in large screening studies. Further studies to improve the generalization of models across a variety of imaging modalities, utilise hybrid or ensemble models, and introduce explainable AI systems with visual and textual interpretability to strengthen clinical trust and adoption should be considered in future research.

## 6 Conclusion

This paper examined the use of deep learning models in the automated detection of retinal diseases, namely Diabetic Retinopathy, Cataracts, and Glaucoma, within the scope of analyzing complete retinal images. It has conducted a comparative evaluation of four convolutional neural network models: a personal Scratch (Tiny VGG) model, EfficientNet-B0, EfficientNet-B7, and AlexNet. Of these, AlexNet achieved the best overall performance across important evaluation criteria, such as accuracy, sensitivity, specificity, precision, negative predictive value, and F-score, and is appropriate for ensuring the reliable use of diagnostic assistance. In addition to the performance measures that remained constant, five-fold cross-validation was also employed in the study to evaluate the model's robustness and generalization. The findings showed a mean $R^2$ of 0.8891 with a low standard deviation of 0.0160 for AlexNet, indicating consistent performance across data partitions and little overfitting. Fold-wise stability also shows that the model can retain diagnostic stability across different data distributions, supporting its suitability for implementation in real-world scenarios. The analysis of computational efficiency also provided additional insights into the model's scalability: AlexNet was the most accurate, with relatively low processing time (around 14 minutes), while deeper networks such as EfficientNet-B7 had longer inference times. This analysis highlights the trade-off between diagnostic accuracy and computational feasibility, an important consideration that should be integrated into time-sensitive clinical settings. SHapley Additive exPlanations (SHAP) were used to interpret the model's decision rationale and improve interpretability. The explanation maps indicated that AlexNet predicts using anatomically significant retinal areas- optic disc and macular areas. This correspondence between model

attention and clinical characteristics increases the transparency and helps clinicians to be confident in AI-assisted diagnostic findings. Together, it can be seen that deep learning models, especially AlexNet, can be used to identify and classify retinal pathologies with high effectiveness, while remaining computationally efficient and interpretable. Quantitative validation combined with computational and explainable AI provides a comprehensive architecture for building reliable diagnostic systems. Future studies are necessary to increase the diversity of datasets, add multimodal imaging modalities (e.g., OCT and fundus fusion), and maximize lightweight yet explainable structures for real-time clinical applications. To summarize, this work contributed to the dynamic subdivision of AI-based ophthalmic imaging, combining high performance with transparency, which is the path to a clinically reliable, explanatory, and scalable diagnostic vision health system.

## Author contributions

**Conceptualization:** El-Sayed M. Elkenawy, Khaled Sh. Gaber, Amel Ali Alhussan, Doaa Sami Khafaga.

**Data curation:** El-Sayed M. Elkenawy, Khaled Sh. Gaber, Ehsan Khodadadi, Amel Ali Alhussan, Doaa Sami Khafaga, Marwa M. Eid.

**Formal analysis:** El-Sayed M. Elkenawy, Khaled Sh. Gaber, Ehsan Khodadadi, Amel Ali Alhussan, Doaa Sami Khafaga, Marwa M. Eid.

**Funding acquisition:** Nima Khodadadi.

**Investigation:** El-Sayed M. Elkenawy, Nima Khodadadi, Khaled Sh. Gaber, Ehsan Khodadadi, Doaa Sami Khafaga, Marwa M. Eid.

**Methodology:** El-Sayed M. Elkenawy.

**Resources:** Marwa M. Eid.

**Software:** Nima Khodadadi.

**Supervision:** Nima Khodadadi.

**Validation:** Nima Khodadadi.

**Visualization:** Nima Khodadadi, Ehsan Khodadadi, Amel Ali Alhussan.

**Writing – original draft:** El-Sayed M. Elkenawy, Khaled Sh. Gaber, Ehsan Khodadadi, Amel Ali Alhussan, Doaa Sami Khafaga, Marwa M. Eid.

**Writing – review & editing:** Nima Khodadadi.

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
