## [Decision Letter · Decision Letter 0]

20 Oct 2025

PONE-D-25-51766Automated Retinal Disease Classification Using Deep Learning and AlexNet with Statistical Models AnalysisPLOS ONE

Dear Dr. Khodadadi,

Thank you for submitting your manuscript to PLOS ONE. After careful consideration, we feel that it has merit but does not fully meet PLOS ONE’s publication criteria as it currently stands. Therefore, we invite you to submit a revised version of the manuscript that addresses the points raised during the review process.

We look forward to receiving your revised manuscript.

Kind regards,

Ali Mohammad Alqudah

Academic Editor

PLOS ONE

Journal Requirements:

Reviewers' comments:

Reviewer's Responses to Questions

**Comments to the Author**

1. Is the manuscript technically sound, and do the data support the conclusions?

Reviewer #1: Yes

Reviewer #2: Yes

2. Has the statistical analysis been performed appropriately and rigorously? 

Reviewer #1: Yes

Reviewer #2: Yes

3. Have the authors made all data underlying the findings in their manuscript fully available?

Reviewer #1: Yes

Reviewer #2: Yes

4. Is the manuscript presented in an intelligible fashion and written in standard English?

Reviewer #1: Yes

Reviewer #2: Yes

5. Review Comments to the Author

Reviewer #1: The abstract should briefly describe the motivation for choosing AlexNet and EfficientNet, emphasizing their architectural differences and rationale for comparison.

Include the key statistical validation (e.g., Wilcoxon signed-rank test) results concisely to demonstrate analytical rigor.

Provide clearer motivation linking clinical significance (e.g., blindness prevention statistics) to computational necessity.

Emphasize the research gap more explicitly — for instance, specify limitations in previous multi-class models or explain why AlexNet still performs competitively despite its older architecture.

Include more recent epidemiological data (2024 WHO Vision Report) for credibility.

Add a critical comparative discussion rather than descriptive summaries in Table 1. Highlight methodological gaps and performance limitations of past works.

Integrate transformer-based or hybrid vision architectures (e.g., Vision Transformers, ConvNeXt) to position the study within current 2023–2025 research trends.

Clarify citation redundancy (e.g., ref [9] and [16] appear repeated).

Clarify hardware configuration (GPU type, memory, training time) and optimizer parameters (learning rate, batch size, epochs).

Include confusion matrices for all models (currently only partially referenced).

Report confidence intervals or statistical significance across cross-validation folds.

Incorporate ROC and AUC metrics for a more comprehensive diagnostic evaluation.

Reviewer #2: Add k-fold cross-validation: Use 5-fold or 10-fold cross-validation to verify generalizability and minimize dataset bias. This would strengthen the statistical robustness of the reported metrics.

Report training time, number of parameters, and FLOPs (Floating Point Operations) for each model to highlight trade-offs between performance and efficiency.

Apply visualization methods such as Grad-CAM, LIME, or SHAP to show how models localize disease regions on the retina. This bridges the gap between AI prediction and clinical interpretability.

Include error analysis:

Analyze misclassified samples to identify patterns (e.g., overlapping features between cataract and DR). This helps justify model limitations and guides dataset improvement.

Deepen the Related Works section:

Incorporate recent (2023–2025) literature on transformer-based retinal models, attention mechanisms, and multimodal learning.

Discuss challenges such as model interpretability, hardware constraints in clinics, and regulatory compliance for medical AI systems.

6. PLOS authors have the option to publish the peer review history of their article (what does this mean?). If published, this will include your full peer review and any attached files.

Reviewer #1: No

Reviewer #2: No

---

## [Decision Letter · Decision Letter 1]

23 Nov 2025

Automated Retinal Disease Classification Using Deep Learning and AlexNet with Statistical Models Analysis

PONE-D-25-51766R1

Dear Dr. Khodadadi,

We’re pleased to inform you that your manuscript has been judged scientifically suitable for publication and will be formally accepted for publication once it meets all outstanding technical requirements.

Kind regards,

Ali Mohammad Alqudah

Academic Editor

PLOS ONE

Additional Editor Comments (optional):

Reviewers' comments:

Reviewer's Responses to Questions

**Comments to the Author**

1. If the authors have adequately addressed your comments raised in a previous round of review and you feel that this manuscript is now acceptable for publication, you may indicate that here to bypass the “Comments to the Author” section, enter your conflict of interest statement in the “Confidential to Editor” section, and submit your "Accept" recommendation.

Reviewer #1: All comments have been addressed

Reviewer #2: All comments have been addressed

2. Is the manuscript technically sound, and do the data support the conclusions?

Reviewer #1: Yes

Reviewer #2: Yes

3. Has the statistical analysis been performed appropriately and rigorously? 

Reviewer #1: Yes

Reviewer #2: Yes

4. Have the authors made all data underlying the findings in their manuscript fully available?

Reviewer #1: Yes

Reviewer #2: Yes

5. Is the manuscript presented in an intelligible fashion and written in standard English?

Reviewer #1: Yes

Reviewer #2: Yes

6. Review Comments to the Author

Reviewer #1: (No Response)

Reviewer #2: (No Response)

7. PLOS authors have the option to publish the peer review history of their article (what does this mean?). If published, this will include your full peer review and any attached files.

Reviewer #1: No

Reviewer #2: No

---

## [Editor Report · Acceptance letter]

PONE-D-25-51766R1

PLOS One

Dear Dr. Khodadadi,

I'm pleased to inform you that your manuscript has been deemed suitable for publication in PLOS One. Congratulations! Your manuscript is now being handed over to our production team.

Kind regards,

on behalf of

Dr. Ali Mohammad Alqudah

Academic Editor

PLOS One